# Indicators of Attack Failure: Debugging and Improving Optimization of Adversarial Examples

**Maura Pintor** [1 2]  **Luca Demetrio** [1]  **Angelo Sotgiu** [1 2]  **Giovanni Manca** [1]  **Ambra Demontis** [1]  **Nicholas Carlini** [3]
**Battista Biggio** [1 2]  **Fabio Roli** [1 2]

## Abstract

Evaluating robustness of machine-learning models to adversarial examples is a challenging problem. Many defenses have been shown to provide a false sense of security by causing gradient-based attacks to fail, and they have been broken under more rigorous evaluations. Although guidelines and best practices have been suggested to improve current adversarial robustness evaluations, the lack of automatic testing and debugging tools makes it difficult to apply these recommendations in a systematic manner. In this work, we overcome these limitations by (i) defining a set of quantitative indicators which unveil common failures in the optimization of gradient-based attacks, and (ii) proposing specific mitigation strategies within a systematic evaluation protocol. Our extensive experimental analysis shows that the proposed indicators of failure can be used to visualize, debug and improve current adversarial robustness evaluations, providing a first concrete step towards automatizing and systematizing current adversarial robustness evaluations.

## 1. Introduction

Neural networks are now deployed in settings where it is important that they behave reliably and robustly (McDaniel et al., 2016; Finlayson et al., 2019; Yuan et al., 2019; Biggio & Roli, 2018). Unfortunately, these systems are vulnerable to *adversarial examples* (Szegedy et al., 2014; Biggio et al., 2013), i.e., inputs intentionally crafted to mislead machine-learning classifiers at test time. This vulnerability has caused a strong reaction from the community, with many proposed defenses (Yuan et al., 2019; Papernot et al., 2016;

---
[*]Equal contribution  [1]Department of Electrical and Electronic Engineering, University of Cagliari, Italy [2]Pluribus One [3]Google. Correspondence to: Maura Pintor <maura.pintor@unica.it>.

*Accepted by the ICML 2021 workshop on A Blessing in Disguise: The Prospects and Perils of Adversarial Machine Learning.* Copyright 2021 by the author(s).

Xiao et al., 2020; Roth et al., 2019). Early defenses often argued robustness by showing the defense could prevent prior attacks, but not attacks tailored to that particular defense. In particular, most attempted defenses to adversarial examples only succeed at increasing the difficulty of solving the minimization formulation, and *not* at actually increasing the robustness of the classifier (i.e., increasing the distance of the decision boundary from the samples) (Carlini & Wagner, 2017a;b; Athalye et al., 2018; Tramer et al., 2020).

Adversarial examples are typically generated through *gradient descent*: the adversary constructs a *loss function* so that a minimum for that function is an adversarial example. While gradient-based attacks are highly effective at finding adversarial examples on undefended classifiers with smooth loss functions, many defenses substantially hinder the attack optimization by obfuscating gradients or by exhibiting harder-to-optimize loss functions. Moreover, even though guidelines and best practices have been suggested to improve current adversarial robustness evaluations, the lack of automatic testing and debugging tools makes it difficult to apply these recommendations in a systematic manner.

**We make the following contributions:** (i) we introduce a unified attack framework that captures the predominant styles of existing gradient-based attack methods, and allows us to categorize the five main causes of failure (Sect. 2); (ii) we propose five *indicators of attack failures* (IoAF), i.e., metrics and principles that help understand why and when gradient-based attack algorithms fail (Sect. 3); (iii) we empirically evaluate the utility of our metrics on four recently-published defenses (Sect. 4; and (iv) we provide open-source code and data we used in this paper (`https://github.com/pralab/IndicatorsOfAttackFailure`). We conclude by discussing limitations and future research directions (Sect. 5).

## 2. Gradient-based Attacks and Failures

We argue here that optimizing adversarial examples amounts to solving a multi-objective optimization:

$$\min_{\boldsymbol{\delta} \in \Delta} \left( L(\boldsymbol{x} + \boldsymbol{\delta}, y; \boldsymbol{\theta}), \|\boldsymbol{\delta}\|_p \right), \tag{1}$$

**Algorithm 1:** Our framework for computing adversarial attacks

**Input** : $\boldsymbol{x}$, the initial point; $y$, the true class of the initial point; $n$, the number of iterations; $\alpha$, the learning rate; $f$, the target model; $\Delta$, the considered region.

**Output**: $\boldsymbol{x}^{\star}$, the solution found by the algorithm

1 $\boldsymbol{x}_0 \leftarrow$ `initialize`$(\boldsymbol{x})$
2 $\hat{\boldsymbol{\theta}} \leftarrow$ `approximation`$(\boldsymbol{\theta})$
3 $\boldsymbol{\delta}_0 \leftarrow \boldsymbol{0}$
4 **for** $i \in [1, n]$ **do**
5     $\boldsymbol{\delta}' \leftarrow \boldsymbol{\delta}_i - \alpha \nabla_{\boldsymbol{x}_i} L(\boldsymbol{x}_0 + \boldsymbol{\delta}_i, y; \hat{\boldsymbol{\theta}})$
6     $\boldsymbol{\delta}_{i+1} \leftarrow$ `apply-constraints`$(\boldsymbol{x}_0, \boldsymbol{\delta}', \Delta)$
7 $\boldsymbol{\delta}^{\star} \leftarrow$ `best`$(\boldsymbol{\delta}_0, ..., \boldsymbol{\delta}_n)$
8 **return** $\boldsymbol{\delta}^{\star}$

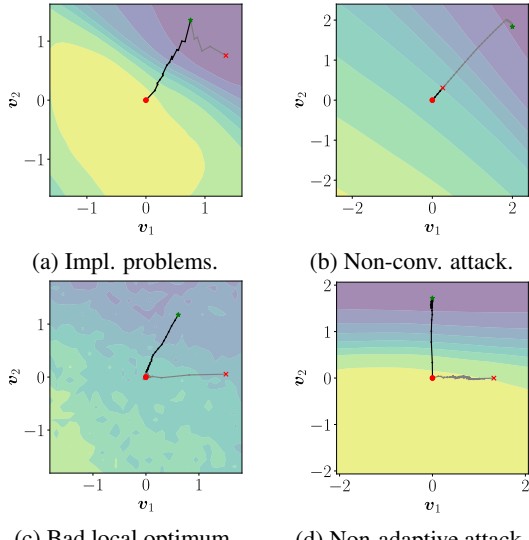

(a) Impl. problems.     (b) Non-conv. attack.

(c) Bad local optimum.     (d) Non-adaptive attack.

*Figure 1.* The four failures that can be encountered during the optimization of an attack. The failed attack path is shown in *gray*, the successful attack is displayed in *black*. The point $\boldsymbol{x}_0$ is marked with the *red* dot, the returned point of the failed attack with a *red* cross, and the successful adversarial point with the *green* star. The loss landscape is represented as $L(\boldsymbol{x} + a\boldsymbol{v}_1 + b\boldsymbol{v}_2, y_i; \boldsymbol{\theta})$. $\boldsymbol{v}_1$ is the normalized direction $(\boldsymbol{x}_n - \boldsymbol{x}_0)$, while $\boldsymbol{v}_2$ is a representative direction for the displayed case.

where $\boldsymbol{x} \in [0,1]^d$ is the input sample, $y \in \{1, \ldots, c\}$ is either its label (for untargeted attacks) or the label of the target class (for targeted attacks), and $\boldsymbol{\delta} \in \Delta$ is the perturbation optimized to have the perturbed sample $\boldsymbol{x}' = \boldsymbol{x} + \boldsymbol{\delta}$ misclassified as desired, within the given input domain. The target model is parameterized by $\boldsymbol{\theta}$. The given problem presents an inherent tradeoff: minimizing $L$ amounts to finding an adversarial example with large misclassification confidence and perturbation size, while minimizing $\|\boldsymbol{\delta}\|_p$ penalizes larger perturbations (in the given $\ell_p$ norm) at the expense of decreasing misclassification confidence.[1] Typically the attacker loss $L$ is defined as the Cross-Entropy (CE) loss, or the logit difference (Carlini & Wagner, 2017b).

Adversarial attacks often need to use an approximation $\hat{\boldsymbol{\theta}}$ of the target model, since the latter may be either non-differentiable, or not sufficiently smooth (Athalye et al., 2018), hindering the gradient-based attack optimization process. In this case, once the attacker loss has been optimized on the surrogate model $\hat{\boldsymbol{\theta}}$, the attack is considered successful if it evades the target model $\boldsymbol{\theta}$.

**Attack Algorithm.** According to the previous discussion, all gradient-based adversarial attacks can be seen as searches for solutions to Eq. 1. Thus, their main steps can be summarized as detailed in Algorithm 1: (i) an *initialization point* (line 1) needs to be set, and this can be done by directly using the input point $\boldsymbol{x}$, or a different point; (ii) the attacker might have to chose a surrogate model $\hat{\boldsymbol{\theta}}$ that approximates the real target $\boldsymbol{\theta}$ (line 2); (iii) the attack iteratively updates the initial point searching for a better adversarial example (line 4), computing in gradient descent steps (line 5); (iv) the

perturbation $\boldsymbol{\delta}_{i+1}$ is obtained by enforcing the constraints defined in Eq. (line 6); (v) at the end of the iterations, the attacker has to select the solution among the perturbations collected along the iterations, formalized as the *attack path* (line 7).

### 2.1. Attack failures

We can now isolate four failures that can be encountered while optimizing adversarial attacks using Algorithm 1, and we bound each of them to specific steps of such procedure. $F_1$: **Implementation Problems.** It might be possible that the used implementation include errors or bugs. $F_2$: **Non-converging attack.** Attacks sometimes do not converge to any local minimum, as shown in Fig. 1b. This can be caused by either the step size, i.e., the algorithm is not exploring the space, or the number of steps of the attack, i.e., the optimization is stopped too early (respectively parameters $\alpha$ and $n$ Algorithm 1, line 5 and line 4). $F_3$: **Bad local optimum.** The attack might reach a region where there is little or useless information to exploit, as shown in Fig. 1c. This might happen because of gradient obfuscation, i.e. the gradients are (nearly) zero (i.e. flat regions), or noisy (line 5 of Algorithm 1). $F_4$: **Non-adaptive attack.** The loss function that the attacker optimizes does not match the actual loss of the target system, and this is caused by a bad choice of the

---

[1]Note that the sign of $L$ may be adjusted internally in our formulation to properly account for both untargeted and targeted attacks.

surrogate model (line 2 of Algorithm 1), as shown in Fig. 1d.

## 3. Indicators of Attack Failure

In this section we describe our Indicators of Attack Failures, i.e. tests that help an analyst debug a failing attack. Each of these tests outputs a value bounded between 0 and 1, where values towards 1 implies the presence of the failure described by the test.

$I_1$: **Silent Success.** This indicator is designed as a binary flag that triggers when the attack is failing, but a legitimate adversarial example is found inside the attack path, as described by the implementation problem failure ($F_1$).

$I_2$: **Break-point angle.** This indicator is designed to quantify the non-convergence of the attack ($F_2$) caused by poor choice of hyperparameters. We normalize the loss along the attack path, and we draw a triangle whose vertices are the first and last point in the loss curve, and the point further to that segment, defining angle $\beta$ (a figure showing how angle $\beta$ is obtained can be found in the Appendix, as Fig. 3a). We measure $1 - |cos\beta|$: when $\beta \approx \pi$, the triangle is flat, i.e. the loss is still decreasing; when $\beta \approx \frac{\pi}{2}$ the loss has not yet reached convergence.

$I_3$: **Increasing loss.** This indicator is designed to quantify either the non-convergence of the attack ($F_2$), or the inability of converging to a good local optimum ($F_3$), both caused by the presence of noisy gradients, where the loss of the attack is increasing while optimizing. We normalize the loss of the attack and the iterations as we did in $I_2$, we extract from it only the portions where it increases, and we sum its area (a figurative example of such metric can be found in the Appendix, as Fig. 3b).

$I_4$: **Zero gradients.** This indicator is designed to quantify the bad-local optimum failure ($F_3$), caused by the absence of gradient information. We compute how many times, along the attack path, the gradients of the loss function are zero. This indicator is close to 1 when most of the norms of the gradient are 0, causing the attack step to fail.

$I_5$: **Non-transferability.** This indicator detects the non-adaptive failure ($F_4$), by measuring if the optimized attack fails against the real target model, while succeeding against the surrogate one. If the attack transfers successfully, the indicator is set to 0, otherwise it is set to 1.

### 3.1. Mitigate the Failures of Security Evaluations

Once the robust accuracy of a model has been computed, the attacker should now check the feedback of the indicators and mitigate accordingly the detected failures.

$M_1$: **Fix the implementation.** If $I_1$ is active, the attack is considered failed, but there exists an adversarial point inside the computed path that satisfies the attack objective. The implementation should be changed.

*Table 1.* Robust accuracies (%) after patching the security evaluations with the prescribed mitigations.

| Model | Rob. acc. |
|---|---|
| *k-WTA* (Xiao et al., 2020) | 58% |
| $M_1 \rightarrow$ | 36% |
| $M_3 \rightarrow$ | **6%** |
| *Distillation* (Papernot et al., 2016) | 94% |
| $M_3 \rightarrow$ | **0%** |
| *Ens. Div.* (Pang et al., 2019) | 38% |
| $M_1 \rightarrow$ | 36% |
| $M_2 \rightarrow$ | **9%** |
| *TWS (Yu et al., 2019)* | 35% |
| $M_5 \rightarrow$ | **0%** |

$M_2$: **Tune the hyperparameters.** If $I_2$ activates, it means that the optimization can be improved, and hence both the step size and iteration hyperparameters can be increased. Otherwise, if $I_3$ activates, the attack should consider a smaller step, as it might be overshooting local minima.

$M_3$: **Use a different loss function.** If $I_3$ activates, and the decrement of the step size did not work, the attack should change the loss to be optimized (Tramer et al., 2020), preferring one that has a smoother behavior. If $I_4$ activates, the attack should consider loss functions that do not saturate (e.g. avoid the softmax) (Carlini & Wagner, 2016), or increase the step size to avoid regions with zero gradients.

$M_4$: **Consider different restarts for the attack.** If $I_3$ or $I_4$ activates, the attack might also consider to repeat the experiments with more initialization points and restarts, as the failure could be the result of added randomness or an unlucky initialization.

$M_5$: **Perform adaptive attacks.** Lastly, if none of the above applied, the attack might be optimizing against a bad surrogate model. If $I_5$ is active, the attack should be repeated by changing the surrogate to better approximate the target, or include the defense inside the attack itself (Tramer et al., 2020). This implies repeating the evaluation, as such change might trigger other previously-fixed failures.

When attacks fail even after the applications of these mitigations, the designer of the defense should try as hard as possible to break the proposed defense with further investigations (Carlini et al., 2019).

## 4. Experiments

We now show the correlation between the feedback of our indicators, and the false sense of security given by badly-evaluated defenses. We apply the following pipeline: (i) we test the defense with the original attack strategy proposed by the author of the defense; (ii) we select the failure cases and inspect the feedback of our indicators *per-sample*; (iii) for each cause of failure, we apply the specific remediation sug-

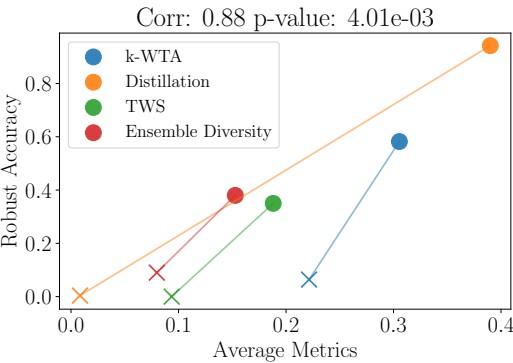

*Figure 2.* Evaluation of our metrics for different models. Robust accuracy vs. average value of the indicators, for the *initial evaluation* (denoted with '○'), with the evaluation *after-mitigation* (denoted with '×')

gested by the metric; and (iv) we show that the attack now succeeds, thus reducing the robust accuracy of the target model, and also the values of the indicators. We select four defenses that have been reported as failing, and we show that our indicators would have detected such evaluation errors, reporting the results of the process in Table 1. Each evaluation was conducted with 5 random restarts.

*k-Winners-Take-All (kWTA)*, proposed by Xiao et al. (2020) uses only the top-k outputs from each layer, generating many discontinuities in the loss landscape, and hence resulting in failure $F_2$. For many failing attacks, the $I_1$ indicator triggers, implying that the attack found an adversarial example inside the path. We then apply mitigation $M_1$, and we lower accordingly the robust accuracy of the model to 36,4%. We analyze the feedback of $I_3$, for inspecting the presence of noisy gradients. We apply mitigation $M_3$, and we change the loss of the attack as described by Tramer et al. (2020). This attack averages the gradient of each single point of the attack path with the information of the surrounding ones. The resulting direction is then able to correctly descent toward a minimum. After such mitigation, the robust accuracy drops to 6.4%, and so follows the indicator.

*Distillation*, proposed by Papernot et al. (2016), works by training a model to have zero gradients around the training points, leading gradient-based attacks towards $F_3$. All the attacks fail because of the absence of gradient information in the cross-entropy loss used by the attacks, leading to bad local optima ($F_3$), and such is highlighted by the feedback of $I_3$. We apply mitigation $M_3$, and we change the loss optimized during the attack, following the strategy applied by Carlini & Wagner (2016), into the logit of the model rather than the final softmax layer. We repeat the PGD attack with such fix, and the robust accuracy drops to 0%, along with the indicator $I_3$.

*Ensemble diversity*, proposed by Pang et al. (2019), is composed with different neural networks, trained with a regularizer that encourages diversity. Firstly, $I_1$ highlighted the presence of $F_1$, implying that some failing attacks are due to the implementation itself. We apply mitigation $M_1$, and the robust accuracy decreases to 36%. Also, $I_2$ is active, implying that the loss of of failing attacks could be optimized more. For this reason, we apply mitigation $M_2$, and we increase the step size to 0.05 and the iterations to 50. This patch lowers the robust accuracy to 9%.

*Turning a Weakness into a Strenght (TWS)*, proposed by Yu et al. (2019), measures how much the decision changes locally around samples to detect adversarial attacks. We consider only part of this defense, as we wish to show that attacks optimized neglecting such term will trigger the non-adaptive failure ($F_4$). The detector is rejecting adversarial attacks successfully computed on the undefended model, triggering the $I_5$ indicator. Hence we apply mitigation $M_5$, and we adapt the attack to consider also the rejection class. This version of PGD minimizes the usual loss function of the attacker, but it also minimizes the score of the rejection class when encountered, allowing it to evade the rejection. We run the attack, obtaining a new robust accuracy of 0%.

As a additional analysis, we want to understand if our indicators are correlated with faults of the security evaluations of defenses. Each original evaluation is characterized by high values of one or more indicator, while the opposite happens for stronger attacks. To gain a quantitative evaluation of out hypothesis, we compute both the p-value and the correlation between the average score of the indicators and the robust accuracy, depicting this result in Fig. 2. Both p-value and correlation suggest a strong connection between these analyzed quantities, confirming our initial belief.

## 5. Conclusions

We propose the use of Indicators of Attack Failure (IoAF), quantitative tests that help debug faulty-conducted security evaluations, and fix them through the systematic application of specific mitigations. We select defenses that have been previously shown to be weak against adversarial attacks, and we evaluate them with the lens of our indicators, showing that their misconduct could have been detected easily in advance. We empirically prove that the indicators are correlated with overestimated robust accuracies, while their values drop when attacks are correctly performed. We acknowledge that we do not provide a fully-autonomous pipeline for adapting attack to an existing defense, but rather helping the adaption to existing attacks by the usage of our work. As future work, we envision the development of interactive dashboards, that can be inspected while debugging the attack. Also, we would like to include our indicators inside the results of other benchmarks (Croce et al., 2020).

## Acknowledgement

This work has been partly supported by the PRIN 2017 project RexLearn (grant no. 2017TWNMH2), funded by the Italian Ministry of Education, University and Research; and by BMK, BMDW, and the Province of Upper Austria in the frame of the COMET Programme managed by FFG in the COMET Module S3AI.

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

# Appendix

## A. Hyperparameters for the attacks

*k-WTA*: We test the robustness of this model by attacking it with $\ell_\infty$-PGD (Madry et al., 2018) with a step size of $\alpha = 0.003$, maximum perturbation $\epsilon = 8/255$ and 50 iterations, with 5 restarts for each attack, scoring a robust accuracy of 58% on 100 samples.

In the patched attack, we run $\ell_\infty$-PGD with the same parameters, but smoothing the gradients by averaging 100 neighboring points from a normal distribution $\mathcal{N}(\mu = \boldsymbol{x}_i, \sigma = 0.031)$, where $x_i$ is a point in the attack path. *Distillation*: We apply, to a model trained on MNIST, $\ell_\infty$-PGD, with step size $\alpha = 0.01$, maximum perturbation $\epsilon = 0.3$ for 50 iterations on 100 samples, resulting in a robust accuracy of 94,2%.

*Ensemble Diversity*: We apply $\ell_\infty$-PGD, with step size $\alpha = 0.001$, maximum perturbation $\epsilon = 0.01$ for 10 iterations on 100 samples, resulting in a robust accuracy of 38%.

*TWS*: We attack a ResNet model defended with such mechanism with $\ell_\infty$-PGD, with step size $\alpha = 0.1$, maximum perturbation $\epsilon = 0.3$ for 50 iterations on 100 samples, and then we query the defended model with all the computed adversarial examples. While the attacks works against the standard model, some of them are rejected by the defense, resulting in a robust accuracy of 35%, highlighted by the trigger of the $I_5$ indicator.

## B. Details on the indicators

We report here graphical examples that can help the reader to fully understand indicator $I_2$ (Fig. 3a and $I_3$ (Fig. 3b).

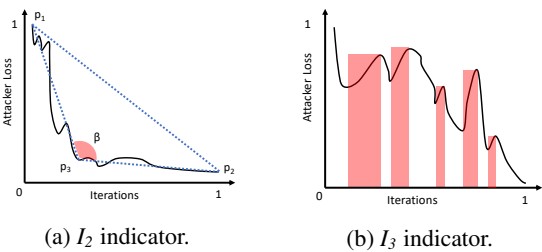

(a) $I_2$ indicator.      (b) $I_3$ indicator.

## C. Additional results

**Values of the indicators** We report in Table 2 the mean values of all the indicators over 100 samples, computed on each selected attack, against the selected defenses. We report also the results of the version of AutoPGD (APGD) (Croce & Hein, 2020) that uses the difference of logit (DLR) as a loss to optimize. This strategy will take care to automatically tune its hyperparameters while optimizing, reducing

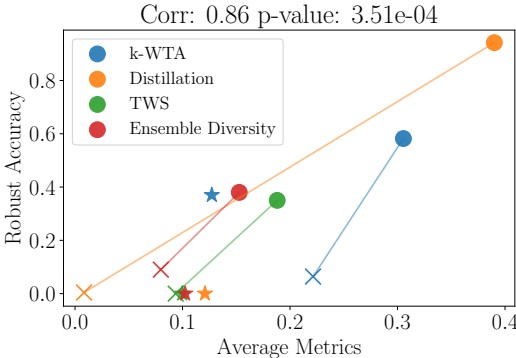

*Figure 4.* Evaluation of our metrics for different models. Robust accuracy vs. average value of the indicators, for the *initial evaluation* (denoted with '○'), with the evaluation *after-mitigation* (denoted with '×'), and the evaluation with APGD (denoted with '★')

possible errors that occur while deciding the values of step size, and iterations. We also report the mean values of the indicators, showing that such is correlated with the robust accuracy of the analyzed model.

**Visualizing the effects of adaptive attacks** We report in Fig. 5 the mean values of each indicators for the original attack, and the adaptive one. We highlight that the latter decreases the values of the indicators, hinting their correlation with the failures of gradient-based attacks. To further show such correlation, we also report the effectiveness of AutoAttack DLR in Fig. 4, where we observe a similar pattern as the one described before.

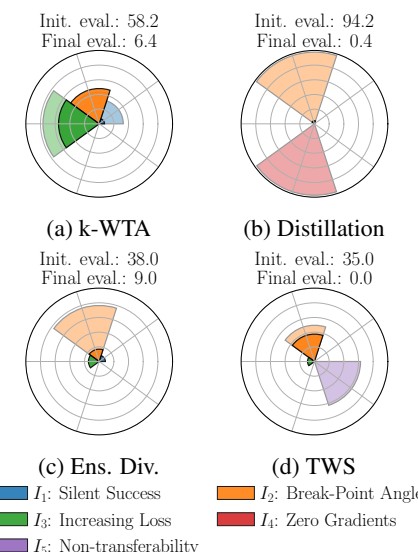

*Figure 5.* The values of our indicators and the success rate (SR) of the attack, before (semi-transparent colored area) and after (solid colored area) fixing the failures, computed for the analyzed models.

| Model | Attack | $I_1$ | $I_2$ | $I_3$ | $I_4$ | $I_5$ | $\bar{I}$ | RA |
|---|---|---|---|---|---|---|---|---|
| | PGD | 0.33 | 0.43 | 0.77 | - | - | 0.306 | 58% |
| *k-WTA* (Xiao et al., 2020) | APGD | - | 0.31 | 0.33 | - | - | 0.128 | 36% |
| | PGD$^\star$ | 0.07 | 0.48 | 0.55 | - | - | 0.220 | 6% |
| | PGD | - | 0.98 | - | 0.97 | - | 0.39 | 94% |
| *Distillation* (Papernot et al., 2016) | APGD | - | 0.40 | 0.21 | - | - | 0.122 | 0% |
| | PGD$^\star$ | - | 0.04 | - | - | - | 0.008 | 0% |
| | PGD | - | 0.76 | - | - | - | 0.152 | 38% |
| *Ensemble Div.* (Pang et al., 2019) | APGD | - | 0.37 | 0.14 | - | - | 0.102 | 0% |
| | PGD$^\star$ | 0.08 | 0.17 | 0.15 | - | - | 0.080 | 9% |
| | PGD | - | 0.49 | 0.07 | - | 0.37 | 0.186 | 35% |
| *TWS* (Yu et al., 2019) | APGD | - | 0.41 | 0.09 | - | - | 0.100 | 0% |
| | PGD$^\star$ | - | 0.37 | 0.10 | - | - | 0.094 | 0% |

*Table 2.* Values of the Indicators of Attack Failures, computed for all the attacks against all the evaluated models. We denote the attacks that apply also the mitigations as PGD$^\star$.