# OpenReview forum: "Indicators of Attack Failure: Debugging and Improving Optimization of Adversarial Examples"
_ICML.cc/2021/Workshop/AML — ICML 2021 Workshop AML Poster_

### Official Review · Reviewer_eJig · 2021-06-20
**This work provide several quantitative indicators of attack failures and possible mitigation strategies for them.**

**Rating:** Accept
**Confidence:** 3

**Review:**

Existing works usually provide qualitative tips of attack failure to avoid a false sense of robustness. This work goes one step further and provides five quantitative indicators of attack failures (IoAF), which might help to systematize or even automate the process of checking the correctness of robustness evaluations. The authors also provide several mitigation strategies for these IoAFs. Based on these IoAFs as well as these mitigation strategies, the authors carried out case studies on several existing defense methods and show the effectiveness of the proposed IoAFs and mitigation strategies.

The paper is clear. The idea of quantitative IoAF is novel.

Pros:
1. This work provides quantitative indicators of attack failures, instead of qualitative tips.
2. The case-study experiments show the proposed IoAFs and mitigation strategies work for evaluating the robustness of state-of-the-art defense methods.

Cons:
1. Some of these quantitative indicators are based on heuristics. Though they are shown to be effective in the experiments, a theoretical analysis would make them more convincing.
2. The case study experiments only show that the indicators will detect problems for failure cases. Adding case studies for known-to-success cases would help to demonstrate these indicators won't give false-positive of attack failures.
3. The defense methods included in the experiments were defeated in existing work (https://arxiv.org/pdf/2002.08347.pdf).

---

### Decision · Program_Chairs · 2021-06-21

**Decision:**

Accept (Poster)

**Comment:**

This paper provided five quantitative indicators of attack failures, which might hep to systematize or even automate the process of checking the correctness of robustness evaluations. The paper may be significant to the field.